# Differences in suicide acceptability by farming-related occupation, demographic, and religiosity factors, general social survey 2000–2022

Jeanne M. Ward[1]*, John R. Blosnich[1,2]

1 Suzanne Dworak-Peck School of Social Work, University of Southern California, Los Angeles, California, United States of America, 2 Center for Health Equity Research and Promotion, VA Pittsburgh Healthcare System, Pittsburgh, Pennsylvania, United States of America

* jeannewa@usc.edu

## Abstract

**Data Availability Statement:** The datasets from the General Social Survey are freely and publicly available from NORC at the University of Chicago at https://gss.norc.org.

### Introduction

Suicide acceptability beliefs must be considered when evaluating interventions to prevent suicide, as such beliefs can reveal cultural sanctions associated with suicide and suicidal behaviors and thoughts. Compared to the general US population, farmers/ranchers have an elevated suicide rate, requiring culturally competent interventions. This analysis investigated whether farmers and workers in agricultural-related industries differ from the general U.S. population in suicide acceptability levels.

### Methods

Cross-sectional General Social Survey (GSS) data were combined from years 2000 to 2022. Four yes/no items assessing whether respondents thought an individual should be able to decide to end their life amid four negative life scenarios were used to define suicide acceptability or endorsement. U.S. Census Bureau industry and occupational codes were used to delineate occupations. Age, sex, race, ethnicity, educational level, survey administration year, and religiosity level were covariates for multiple logistic regression analyses. Among 18,191 respondents to the GSS, 167 people worked in farming/ranching roles, including 74 farmers/ranch operators and 93 farm/ranch workers.

### Results

In unadjusted models, individuals in farm-related occupations had a lower prevalence than the general US adult population of sanctioning suicide if facing an incurable disease. Yet, suicide endorsement beliefs showed no statistically significant differences between farmers/ranchers and the general population after demographic factors were included in the model. Age, sex, race, ethnicity, and education were significant predictors of suicide acceptability, $p$ < 0.01. The prevalence of farmer/ranch operators identifying as very or moderately religious was significantly higher than that of farm/ranch workers and the general population, $p$ <

**Funding:** This work was supported by a research award from the National Institutes of Health New Innovators Award and the National Institute of Mental Health (DP2MH129967) to J.R.B. The funders had no role in the study design, data collection and analysis, and decision to publish, or preparation of the manuscript.

**Competing interests:** The authors have declared that no competing interests exist.

0.05. The sample identifying as non-religious had higher odds of sanctioning suicide when faced with an incurable disease (aOR 5.980, 95%CI 5.235–6.829), bankruptcy (aOR 3.281, 95%CI 2.791–3.857), having dishonored their family (aOR 3.215, 95%CI 2.732–3.784), or becoming tired of living (aOR 3.660, 95%CI 3.209–4.175).

## Conclusion

The present results showed that farmers'/ranchers' acceptability of suicide was not distinct from the general US population in multivariable models. However, given their disproportionately high suicide rate, they require customized outreach and interventions. Further research may elucidate how religiosity, demographic factors, and beliefs about suicide and religion impact interventions to prevent suicide for individuals working in farming/ranching.

## Introduction

Suicide has increased 35% since 2001 in the U.S. [1], and it is unequally distributed across the general population, with elevated rates occurring among different demographic characteristics, including occupation. Occupation has traditionally been operationalized in suicide research as formal employment for wages, which, unsurprisingly, has a strong effect on mental health because occupation directly affects financial stability [2,3]. A substantial literature shows that financial stability, notably the loss of or lack of financial stability, has a strong, positive association with suicidal ideation, attempt, and suicide death [4,5]. In addition to financial status, other socio-demographic factors are significantly correlated with suicide, such as age, gender, ethnicity/race, years of education, and religiosity [3,6–8].

One occupational group with elevated rates of suicide is agricultural workers (i.e., farmers/ranchers). In 2016, the suicide rate of males working in agriculture, forestry, fishing, and hunting in 32 U.S. states was 36.1 per 100,000, significantly higher than that among working males, which had a suicide rate of 27.4 per 100,000 [9]. The suicide rate of U.S. male farmers was 46.2 and 37.6 per 100,000 for crop and animal producers, respectively, in 2021; during the same period, the combined agriculture industry (agriculture, forestry, hunting, and fishing) had an incident suicide rate of 47.9 per 100,000 [10]. Rural suicides overall increased by 39.4% between 1999 and 2019 [11]. Farmer/rancher suicides tend to be in individuals over the age of 60, male, widowed, and White when compared to non-farmer populations [11]. More than 60% of farmer suicide decedents did not have previously identified mental health diagnoses [11], and the lack of diagnosed mental health conditions is especially prominent among rural men [12]. Farmers/ranchers have lower health-seeking behaviors than the general US population [13–15], and rural populations are less likely than urban ones to utilize health services [16,17]. In addition to deleterious effects on health, lower access and utilization of health care undercuts suicide research from clinical data sources, which have contributed to progress in suicide prevention for military veterans [18]. Although data are limited, national self-reported surveys offer opportunities to learn about risk factors for suicide among individuals working in farming/ranching occupations.

Farmers/ranchers face a variety of suicide-related stressors; paramount among these are financial problems. Farmers'/ranchers' livelihoods are inextricably linked to unpredictable and uncontrollable environmental forces, such as weather patterns or natural disasters, as well as volatile macroeconomic factors, such as fluctuating commodity prices [19,20]. Furthermore,

the nature of farming/ranching requires these occupations to be predominantly in rural and frontier areas of the country, leading to geographic factors associated with social isolation [20] and lack of access to health care services [13,15]. Farmers/ranchers also face well-documented physical, chemical, and environmental hazards, including chronic respiratory and musculo-skeletal disorders as well as various cancers [13,21,22], which have been linked to occupation-related hazards such as pesticide use or dust exposure [23,24]. The complex web of these risk factors helps to explain the greater rate of suicide among individuals in farming/ranching occupations.

Culturally aligned interventions are necessary among farmers/ranchers to prevent suicide, as suicide expression, methods, and risk factors vary according to rural cultural factors, including fatalism, a sense of independence, and stoicism [25]. Against a backdrop of geographic isolation [26,27], these risk factors complicate outreach to promote mental health [25,28–30]. The identity internalized by farmers as hardened, autonomous, and fiercely independent providers [31,32] shapes their expectations and personal values. Rural farming/ranching populations also face considerable stigma about mental healthcare utilization, and they may have geographic limitations to mental health assistance [13,33,34], all of which may impact suicidal thoughts and behaviors as well as personal beliefs about suicide.

Although suicide is complex and multifactorial, social beliefs about suicide are important to consider for prevention interventions. Beliefs about suicide encompass several components, with some instruments gauging stigmatizing opinions about suicide (e.g., Stigma of Suicide Scale) [35], and other instruments presenting scenarios in which participants indicate if they think suicide would be a potential solution (e.g., "If I found out my partner was having an affair, I might try to kill myself") [36]. This latter concept within beliefs about suicide involves assessing whether individuals believe suicide is an acceptable action, including rationalizing suicidal behavior and whether and how to die by suicide [37,38]. Suicide acceptability is predictive of suicide deaths, increasing its risk by up to two-fold [39]. Suicide acceptability is the endorsement of suicide as an understandable or viable action in the face of a challenge or stressful situation [40]. Feigelman et al. [41] found that historical unemployment and religious attendance did not significantly predict suicide mortalities, yet previous suicide acceptability indicators had a strong association with suicide deaths. Another study found that adolescents and young adults with high suicide acceptability had 13 times the odds of planning suicide, even once the model was adjusted to include other factors [42]. These attitudes may be malleable intervention points to inform the prevention of suicidal behavior in high-risk populations. Some may believe that suicide is a justifiable action to stress or an escape from a hopeless future [43], which aligns with the concept of entrapment in the Integrated Motivational-Volitional model of suicide [44]. Similarly, the Interpersonal Theory of Suicide posits that hopelessness and thwarted belonging precede suicidal ideation [45]. If an individual faces these factors and also believes that suicide is acceptable and inevitable, these issues can become mutually reinforcing, making the individual less likely to seek or adopt coping and prevention behaviors [46].

Suicide acceptability may also be influenced by religiosity. There is no standard operational definition of religiosity because it has several dimensions (e.g., strength of belief, self-rated importance, personal activities such as prayer, public activities such as attendance at services) [47], and is measured in different ways; one study reviewed 177 measures of religiosity [48]. Because most religious denominations include strong messaging that prohibits self-harm [49], it is likely that individuals who endorse greater religiosity would be less likely to view suicide as an acceptable act to any stressor. Religiosity has historically been stronger in rural than urban areas [50], which suggests that farmers/ranchers (who are highly likely to live rural areas) would exhibit stronger religiosity than non-farming/ranching occupations.

Consequently, understanding suicide acceptability for farmers/ranchers should also investigate religiosity.

As a group with a high suicide rate with unique occupational and cultural risk factors, farmer/rancher beliefs about suicide acceptability are essential to understand, yet there is limited research in this area. Whether suicide acceptability would be higher or lower in farmers/ranchers than in the general population is unknown. Further, there may be differences between farm/ranch operators and farm/ranch workers. Despite working in similar environmental settings, farm/ranch operators and farm/ranch workers often have different stressors. For instance, farm/ranch operators consider the financial well-being of the complete farm/ranch and have multigenerational operations [11,51]. At the same time, farm/ranch workers often migrate from other regions to find work and live for months away from their communities, facing unique stressors [11].

Therefore, this study aimed to elucidate differences in social attitudes toward suicide in the general population versus those in the farming/ranching population, further differentiating between farmers/rancher operators and farm/ranch workers. Based on their greater suicide risk and unique population characteristics (e.g., physical nature of work, less access to health care), we hypothesized that individuals in farming/ranching occupations would have more suicide acceptability than the general population. The analysis, using data from the General Social Surveys (GSS), focused on four items asking respondents about suicide acceptability, defined as rationalizing suicide as a viable option when faced with adversity [52]. Studies have found links between higher suicide acceptability/endorsement and suicide rates [41,53], and studies with GSS data have measured this concept in various U.S. populations, including those facing financial and legal hardship [40], veterans [54], sexual minorities [55], Black and African-Americans [56], and population-wide [39].

## Data and methods

### Sample

Data for the analyses are from the combined GSS 1972–2022 made publicly available from the National Opinion Research Center (NORC) at the University of Chicago. Conducted biannually, the GSS includes demographic questions along with items gauging various social behaviors and beliefs and is administered to a probability-based, representative sample of adults over the age of 18 residing in the U.S. [57]. The GSS uses block quota sampling to ensure a representative sample by sex, age, and employment status, and it also conducts sampling based on Census tract information and Standard Metropolitan Statistical Areas or non-metropolitan counties [58]. The GSS is a well-known data source used in approximately 14,000 journal articles over its 50+ year history [57]. More in-depth methodology about the GSS is available through the NORC website (https://gss.norc.org/get-the-data) [59,60]. The GSS questionnaire contains core questions asked of all participants about socio-demographics and topical modules, which are split among respondents regarding attitudes, and behaviors, and topics of special interest, such as crime and moral questions, for an interview of approximately 90-minutes in total [61]. Survey years for the present analysis were limited to 2000–2022 to align with the current era in the U.S. during which the suicide rate has risen [62].

### Independent variable

The GSS categorized respondents by jobs based on 2010 occupation and industry codes derived by the U.S. Bureau of the Census. These occupational codes were used to identify two separate groups: (1) farmers, ranchers, and other agricultural managers; and (2) miscellaneous agriculture workers.

## Dependent variables

Respondents were asked whether they believed someone should be able to make the decision to end their life if faced with four hypothetical situations: "has an incurable disease," "has gone bankrupt," "has dishonored his or her family," or "is tired of living and ready to die." These statements excluded the term "suicide." Still, they map onto theorized stressors that precipitate suicide risk, including hopelessness, perceived burdensomeness and bankruptcy [45,63], and interpersonal guilt and shame [64]. Each scenario was presented as a yes/no question and coded individually into four dependent variables. The reliability of the suicide acceptability items was calculated in previous research as .77 and .75 for the GSS in the 2010s and 1980s, respectively, which is generally considered acceptable [65]. In the present analysis, the Cronbach's alpha of the GSS suicide acceptability scale in the dataset was .77, considered reliable. Because the response options to the suicide acceptability items are dichotomous, we also calculated a Kuder-Richardson (KR-20) statistic, which was .76 and also within a range considered acceptable [66].

## Covariates

Age, race, ethnicity, and education were included due to their covariance with suicide and attitudes toward suicide [42,67,68]. A continuous measure of age, ranging from 18–89, was used, and gender was collected in the survey as a binary measure (i.e., man or woman). Racial identity was coded as Black, Other, White, or missing, and respondents with missing racial identity information were categorized as unknown to be preserved for analyses. A dichotomous measure of Hispanic ethnicity (yes/no) was also included in the analyses. Respondents indicated the number of years of education attained, ranging from 0 to 20. The year of the GSS survey was also included as a covariate and was coded continuously from 2000 to 2022.

Religiosity, commonly operationalized through various measures of strength of religious beliefs (i.e., religious attendance, self-rated importance of religious beliefs), is robustly associated with suicide acceptability [69]. In the GSS, religiosity was asked with one item: "Would you call yourself a strong [insert stated religious preference] or not a very strong [insert stated religious preference]" [59]? Strongly, not very strongly, somewhat strongly, and no religion were the response choices. This question about religiosity was revised in 2021 to the following: "To what extent do you consider yourself a religious person" [59]? Very religious, moderately religious, slightly religious, or not religious at all were the available response choices. Because both items measured religiosity, they were harmonized and recoded into a single variable of religiosity: not religious, slightly or somewhat strongly religious, moderately or not very strongly religious, and very or strongly religious.

## Analyses

The GSS used a complex sampling design, so the included survey weights were used for all analyses to account for the survey design and to generate nationally representative estimates. Bivariate categorical statistical tests (Pearson's chi-square) of demographic variables (i.e., gender, race, ethnicity, and religiosity) and suicide acceptability were used to examine differences between farm/ranch operators, farm/ranch workers, and the general population. Differences in age and years of educational attainment across the groups were ascertained with weighted comparisons of means. To test the study's main hypothesis, multivariable logistic regression was used to adjust for covariates while determining if farming/ranching occupations were positively associated with suicide acceptability outcomes. Regression results are reported as adjusted odds ratios with corresponding 95% confidence intervals. Statistical significance was assessed at $p < .05$. Missing values were handled with listwise deletion. Because the GSS dataset

is de-identified and publicly available, the University of Southern California institutional review board deemed this not human subjects research.

## Results

Among the 18,191 respondents, 167 people worked in farming-/ranching-related occupations: 93 farm/ranch operators and 74 farm/ranch workers. The average age of the sample was 46.97 years, and most individuals were women (53.2%), White (76.0%), and non-Hispanic (87.0%) in the GSS sample general population. The average number of years of education in the general population was 13.73 years (see Table 1). In contrast with the GSS sample, farm/ranch operators were significantly older, more likely to be men, and had fewer years of education than the general US population. Within the farming/ranching group, farm/ranch operators were significantly ($p < .05$) older, more likely to be men, had fewer people identifying as Hispanic, and had more years of education than farm/ranch workers. Individuals in farming/ranching roles also had higher percentages of individuals identifying as strongly or moderately religious compared to the general US adult population, with 37.9% of farm/ranch operators identifying as strongly religious compared to 32% of farm/ranch workers and the general population identifying as strongly religious.

**Table 1. Prevalence and unadjusted differences in socio-demographics and suicide acceptability/endorsement by farming/ranching versus the general population, General Social Survey 2000–2022.**

| | Farm/ranch Operator | | Farm/ranch Worker | | General population | | |
|---|---|---|---|---|---|---|---|
| | (n = 93) | | (n = 74) | | (n = 18,024) | | |
| *Socio-demographics* | n | (%) | n (%) | | n | (%) | *P* |
| Gender | | | | | | | |
| Women | 8 | (11.5) | 33 (31.3) | | 9,883 | (53.2) | < .001 |
| Men | 66 | (88.5) | 60 (68.7) | | 8,141 | (46.8) | |
| Race | | | | | | | |
| Black | 3 | (3.0) | 9 (10.2) | | 2,675 | (13.6) | < .001 |
| Other | 4 | (7.7) | 31 (39.3) | | 1,669 | (10.6) | |
| White | 67 | (89.3) | 53 (50.5) | | 13,683 | (75.8) | |
| Ethnicity | | | | | | | |
| Hispanic | 6 | (9.8) | 41 (49.8) | | 2,118 | (13.2) | < .001 |
| Non-Hispanic | 68 | (90.2) | 52 (50.2) | | 15,909 | (86.8) | |
| Religiosity | | | | | | | |
| Strongly/very religious | 26 | (37.9) | 30 (31.6) | | 5,681 | (31.6) | .03 |
| Moderately/not very strongly religious | 36 | (48.6) | 41 (43.6) | | 6,498 | (36.2) | |
| Slightly/somewhat strongly religious | 7 | (9.2) | 10 (12.3) | | 2,032 | (11.2) | |
| Not religious | 5 | (4.3) | 12 (12.5) | | 3,816 | (20.9) | |
| | M | S.E. | M S.E. | | M | S.E. | |
| Age in years | 61.06 | 1.96 | 50.44 2.46 | | 46.97 | .17 | < .001 |
| Education in years | 11.34 | .52 | 8.67 .56 | | 13.73 | .03 | < .001 |
| *Suicide acceptability/endorsement[a] when*: | | | | | | | |
| Facing incurable disease | 39 | (51.5) | 44 (42.6) | | 11,135 | (63.6) | < .001 |
| Bankrupt | 5 | (6.7) | 4 (4.6) | | 2,005 | (11.0) | .076 |
| Dishonored family | 5 | (7.4) | 6 (9.1) | | 1,951 | (10.6) | .705 |
| Done with living | 11 | (15.4) | 16 (14.7) | | 3,482 | (19.3) | .405 |

Notes: Frequencies are unweighted; percentages, means, and standard errors (S.E.) are weighted.

a = Centered on questions beginning with: "Do you think a person has the right to end their own life if. . ..".

**Table 2. Adjusted odds of suicide acceptability/endorsement by farming versus non-farming occupations, General Social Survey 2000–2022.**

| | Odds of endorsing suicide is acceptable/endorsed when: | | | | | | | |
| --- | --- | --- | --- | --- | --- | --- | --- | --- |
| | Facing incurable disease (n = 17,506) | | Bankrupt (n = 17,883) | | Dishonored family (n = 17,858) | | Done with living (n = 17,641) | |
| Independent Variables | aOR | (95%CI) | aOR | (95%CI) | aOR | (95%CI) | aOR | (95%CI) |
| Occupation | | | | | | | | |
| Farm/ranch operator | 0.816 | (0.478–1.393) | 0.990 | (0.379–2.583) | 1.121 | (0.433–2.900) | 1.066 | (0.501–2.268) |
| Farm/ranch worker | 0.861 | (0.528–1.403) | 0.726 | (0.258–2.046) | 1.615 | (0.541–4.822) | 1.375 | (0.769–2.459) |
| Age (years) | 0.996 | (0.993–0.998)* | 0.988 | (0.984–0.992)* | 0.988 | (0.984–0.991)* | 0.999 | (0.996–1.002) |
| Sex | | | | | | | | |
| Male | Ref | | Ref | | Ref | | Ref | |
| Female | 0.839 | (0.775–0.908)* | 0.816 | (0.724–0.920)* | 0.794 | (0.703–0.897)* | 0.819 | (0.747–0.890)* |
| Race | | | | | | | | |
| White | Ref | | Ref | | Ref | | Ref | |
| Black | 0.413 | (0.365–0.467)* | 0.685 | (0.561–0.836)* | 0.665 | (0.550–0.806)* | 0.661 | (0.568–0.769)* |
| Other | 0.736 | (0.626–0.865)* | 0.645 | (0.517–0.805)* | 0.673 | (0.539–0.840)* | 0.633 | (0.512–0.781)* |
| Ethnicity | | | | | | | | |
| Non-Hispanic | Ref | | Ref | | Ref | | Ref | |
| Hispanic | 0.613 | (0.531–0.708)* | 0.751 | (0.604–0.934)* | 0.717 | (0.578–0.891)* | 0.856 | (0.719–1.019) |
| Religiosity | | | | | | | | |
| Strongly/very religious | Ref | | Ref | | Ref | | Ref | |
| Moderately/not very religious | 2.820 | (2.555–3.112)* | 1.450 | (1.236–1.702)* | 1.402 | (1.199–1.639)* | 1.567 | (1.385–1.773)* |
| Slightly/some-what religious | 2.626 | (2.284–3.018)* | 1.238 | (1.004–1.526)* | 1.263 | (1.023–1.560)* | 1.463 | (1.253–1.709)* |
| Not religious | 5.980 | (5.235–6.829)* | 3.281 | (2.791–3.857)* | 3.215 | (2.732–3.784)* | 3.660 | (3.209–4.175)* |
| Education (years) | 1.118 | (1.101–1.135)* | 1.125 | (1.100–1.151)* | 1.115 | (1.089–1.142)* | 1.104 | (1.084–1.124)* |
| Survey year (years) | 1.011 | (1.005–1.017)* | 1.014 | (1.005–1.023)* | 1.008 | (1.000–1.017) | 1.012 | (1.004–1.019)* |

All models were adjusted for age, race, gender, religiosity level, education, and year of survey administration.

* = $p < .05$.

Generally, farm/ranch operators and farm/ranch workers reported a lower prevalence of acceptability for suicide in all scenarios than individuals in the overall U.S. population. In the unadjusted analyses, the general population and farm/ranch operators showed no significant differences in suicide acceptability/endorsement measures. When faced with the hypothetical incurable disease situation, farm/ranch workers were significantly less likely than the members of the general population to sanction suicide (42.6% vs. 63.6%, respectively, $p < .001$). However, after including covariates, this difference was not statistically significant (aOR = 0.861, 95%CI = 0.528–1.403) (Table 2).

Gender, race, and years of education were significant predictors of all suicide endorsement measures, $p < 0.001$, with women less likely than men to endorse suicide as acceptable. Black and other races were less likely to endorse suicide as acceptable, and Black individuals were 59% less likely than White individuals to endorse suicide when faced with an incurable health diagnosis or disease (aOR = 0.413, 95%CI = 0.365–0.467) (Table 2). Age and ethnicity were significant predictors of all suicide endorsement measures except the scenario when the respondent decides they are done living. Hispanic individuals were less likely than non-Hispanic

individuals to endorse suicide as acceptable in all scenarios, with Hispanic individuals having 39% lower odds of endorsing suicide when confronted with an incurable disease. Years of education was a significant predictor of suicide endorsement in all scenarios, with increasing years of education associated with increased odds of suicide acceptability (aOR range: 1.104–1.125). Religiosity was a significant predictor of suicide acceptability, with those identifying as the least religious having higher odds of endorsing suicide for all scenarios, especially amid the scenario of a terminal disease (aOR = 5.980, 95%CI = 5.235–6.829).

## Discussion

The results did not support the hypothesis that individuals in farming/ranching occupations would have greater suicide acceptability than the general US population in this study of suicide acceptability/endorsement among farmers. Generally, individuals in farming/ranching roles had a lower prevalence of endorsing suicide acceptability, and at least in one scenario (incurable disease), the general US population had a significantly higher unadjusted prevalence than farm/ranch workers. The hypotheses were based on public health data showing a high rate of suicide among people in farming/ranching occupations [9,70], as well as well-documented risk factors identified for this group (e.g., lower access to care, greater social isolation) [13,15,32]. However, results indicated a potentially key protective factor among people in farming/ranching occupations: religiosity. We consider this factor as well as the other key outcomes below.

Although religiosity is defined in several ways (e.g., frequency of service attendance, fervor of beliefs, and importance of faith identity) [71], generally it is protective against suicidal thoughts and behaviors and suicide death [72–74]. The present study results showed that individuals in farming/ranching occupations were significantly more likely to endorse higher levels of religiosity than the general US population, which corroborates other findings regarding farmers' religious tendencies, especially in the face of economic hardship [75,76]. Thus, greater religiosity may explain the lower acceptability of suicide observed in the present study. However, religiosity may still have its limits in protection, given that individuals in farming/ranching occupations have greater rates of suicide death. It is important to note that suicidal ideation, attempt, and death are related but distinct phenomena because not all individuals experiencing suicidal ideation will continue to attempt suicide, and less than 10% will die by suicide after attempting to do so [77]. Unfortunately, items about suicidal ideation or attempt are not included in the GSS, precluding investigating how religiosity and suicide acceptability may be associated with these specific suicidal outcomes. Nonetheless, the present findings suggest that religiosity may be a furtive area for investigating suicide risk and prevention for individuals in farming/ranching occupations.

The results around suicide endorsement in the scenario of terminal disease require more attention against the background of literature around physical health for individuals in farming/ranching occupations. People in farming/ranching industries have a greater prevalence of several physical health problems than the general population [78], including lung disease [21,24], chronic pain, arthritis [22], and several forms of cancers [51], including skin, stomach, and brain [79]. Pesticide usage has been associated among farmers with thyroid cancer [80] and lung cancer [81]. Yet, despite these disparities in chronic conditions and disease, individuals in farming/ranching occupations were generally less likely to endorse suicide in relation to having an incurable disease. One potential explanation may relate back to the aforementioned factor of religiosity, which could increase acceptance of one's condition or quality of life [82–84]. For instance, studies of individuals with cardiovascular disease and cancer have shown that greater religiosity was associated with greater quality of life [85,86], which negatively correlates with suicide acceptability. However, while individuals in farming/ranching roles were

generally less supportive of suicide acceptability when faced with terminal disease when compared to the general US population, a little over half of farm/ranch operators endorsed suicide in the situation of terminal disease. Additional research is needed to understand how suicide acceptability may vary depending on the burden of health conditions experienced by individuals in farming/ranching occupations.

The lack of differences around suicide acceptability in the face of bankruptcy was also unexpected, given the substantial literature framing financial problems as a major risk factor for suicide among farmers/ranchers [87,88]. This may point to a difference in responding to hypothetical scenarios versus facing the actual realities of financial strain. It may also suggest stoicism among farming/ranching [89–91], in which individuals silently and solely bear away hardship or pain. Consequently, individuals in farming/ranching occupations may respond to hypothetical situations of hardship by endorsing only solutions of continual hard work, a hallmark of stoicism. In a study of livestock farmers, Nye et al. highlighted a quote from a participant who described farmers as people who "just keep going and going and going until they drop" [92]. Stoicism is not well understood in the empirical suicide risk literature [93], though it has been associated with key suicidogenic constructs such as thwarted belongingness [94]. Further research is needed to explore how individuals with high stoicism (e.g., farmers/ranchers) may differently engaged or interpret measures about suicide acceptability.

Although occupation differences in suicide acceptability in the multivariable models were not significant, the unadjusted model differences are important, especially amid the disproportionately higher suicide rates faced by individuals in farming/ranching occupations [51]. Farmer/rancher suicide decedents are less likely than suicide decedents in the general population to indicate suicidal ideation [95]. Thus, without these typical warning signs (e.g., suicidal ideation, suicide acceptability), additional research into sentinel correlates of suicide attempts is needed to enhance the detection of risk in farmer/rancher populations. There is potential to investigate the need for farmer-specific interventions, especially when farmers face terminal diseases, and to introduce coping strategies.

Although the study used nationally representative survey data that included religiosity and suicide acceptability, there are several limitations. Because the population of farmers/ranchers is quite small compared to other occupations, few survey respondents indicated occupations as farm/ranch operator or farm/ranch workers, limiting the sample size. Further research with larger sample sizes is necessary to capture the nuances among people identifying as farm/ranch operators and farm/ranch workers. Over half of U.S. farm/ranch operators work in off-farm occupations [96] to generate sufficient income, so many may identify as working in other occupations, including the construction industry (18.7%) or manufacturing (12.9%) [97]. Spouses of farm/ranch operators often work in education (22.3%) or healthcare (19.7%) [97]. Service-economy jobs have been replacing agriculture jobs, growing from 40% of nonmetro jobs in 1970 to 57% in 2019, and farm income is generally less than 20% of total household income [98]. Thus, relying solely on census codes for primary occupations may underestimate the number of individuals who are involved in farming, but perhaps not as their sole form of income.

Furthermore, while the GSS surveys a representative sample of U.S. adults and therefore the estimates should be also nationally representative, the small sample size of farmers/ranchers contributes to instability in these estimates. However, the proportion of individuals working on farms in the sample aligns with that of the general population, 1.2% [99]. Horst & Marion (2019) used a representative United States sample from the National Agricultural Workers Survey (NAWS) from 2013–2014, also categorizing farmers/ranchers into farmworker/laborer and farm operator categories [100]. In the Horst & Marion study (2019), the sample demographic percentages were similar to that of the present study sample for gender among farm/

ranch operators (13.7% female) and farm/ranch laborers/workers (38% female). For farm/ranch operators, the percentage of white (97.1%) and non-Hispanic (93.1%) identified in Horst & Marion was consistent with proportions identified in the present sample from the GSS. In national counts from 2021 [101], 28% of farm/ranch workers were female, and 3% of farm/ranch operators were Black. Consequently, the GSS sample of individuals working in farming/ranching occupations, though small, corroborates demographics documented in other studies with representative samples and national statistics.

This study was consistent with others in its significant socio-demographic correlates of suicide (age, sex, race, ethnicity, education, occupation, religiosity) [2,6,7], and it introduces questions about the factors of occupation and financial issues and their interactions with demographics and religiosity. Demographics are important factors to consider in understanding risk of suicide, as they affect the dynamics of society and the experiences of individuals. These factors are interrelated, especially as demographics affect social and economic standing, which are in turn affected by occupation [102,103], exacerbating further inequity in individual and population health [104]. Further, religiosity affects social norms, values, and beliefs, impacting social cohesiveness and political ideologies, which are associated with suicide acceptability [56,69,105]. These interactions with occupation, demographics, and religion also can exacerbate social problems, influencing the factors of hopelessness and burdensomeness that increase suicidal ideation [106]. For example, unemployment [107], low socioeconomic position [108], discrimination and exclusion based on demographics [109,110], and rigid religious beliefs [111,112] all play roles in the environment and wellbeing of an individual. These are important considerations when understanding how to develop effective interventions and support systems to support mental health and prevent suicide.

Despite the numerous covariates included in these analyses, other demographic and behavioral factors not included in the survey could be correlated with suicide acceptability, such as surviving a suicide loss. The GSS validity for suicide acceptability/endorsement may also be limited because the situations described were hypothetical and did not specifically include "suicide" in the item wording. Different language and response items are used in other suicide acceptability instruments [113]. The GSS does not ask respondents about suicidal thoughts and behaviors, so it was not possible to assess how suicide acceptability was associated with suicidal thoughts or behaviors.

Despite limitations, these data about how suicide acceptability in a sample of individuals working in farming/ranching compared to the U.S. population provide novel information for further research and approaches to message framing targeted toward the farming/ranching population. Future research should look at the nuances of religiosity as related to suicide acceptability and the religious identity of suicide decedents. As more socially versus medically oriented approaches to preventing suicide are considered [114] and amid a less religious population [115], it will be essential to understand views of morality and the intersections of identities with industry.

## Conclusion

After accounting for demographic covariates, people in farm/ranch occupations did not show distinct suicide acceptability levels from that of the general population when faced with adverse life scenarios. Age, sex, race, ethnicity, and education were significant predictors of suicide acceptability in the general population. Lower religiosity levels were associated with greater odds of suicide acceptability. Farmers/ranchers going through adverse life events may have engrained beliefs about suicide acceptability, necessitating customized outreach and interventions. Religiosity may be protective against endorsing suicide as an acceptable action, but it

needs to be examined among farmers/ranchers. Further research is required to determine how these demographic factors and beliefs about suicide acceptability and religion impact interventions to prevent suicide and interact with one another. The elucidation of suicide acceptability and its relationship with other factors may offer additional insights as prevention initiatives are designed around the risk factors of suicide among farmers and ranchers.

## Acknowledgments

Thank you to NORC at the University of Chicago for sharing the GSS data.

## Author Contributions

**Conceptualization:** Jeanne M. Ward, John R. Blosnich.

**Data curation:** John R. Blosnich.

**Formal analysis:** Jeanne M. Ward, John R. Blosnich.

**Investigation:** Jeanne M. Ward.

**Methodology:** Jeanne M. Ward, John R. Blosnich.

**Project administration:** Jeanne M. Ward.

**Software:** John R. Blosnich.

**Supervision:** John R. Blosnich.

**Validation:** Jeanne M. Ward.

**Writing – original draft:** Jeanne M. Ward.

**Writing – review & editing:** Jeanne M. Ward, John R. Blosnich.

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
