## [Decision Letter · Decision Letter 0]

6 May 2024

PMEN-D-24-00103

Differences in Suicide Beliefs by Farming-Related Occupation, Demographic, and Religiosity Factors, General Social Survey 2000-2022

PLOS Mental Health

Dear Dr. Jeanne M. Ward,

Thank you for submitting your manuscript to PLOS Mental Health. After careful consideration, we feel that it has merit but does not fully meet PLOS Mental Health’s publication criteria as it currently stands. Therefore, we invite you to submit a revised version of the manuscript that addresses the points raised during the review process.

Reviewer two specifically points out the need to provide definitions in the introduction to enable readers appreciate the major concepts right from the start. Also re-writing the conclusion. 

Please submit your revised manuscript by June 30, 2022. If you will need more time than this to complete your revisions, please reply to this message or contact the journal office at mentalhealth@plos.org. Please include the following items when submitting your revised manuscript:

We look forward to receiving your revised manuscript.

Kind regards,

Martin Mabunda Baluku, Ph.D.

Academic Editor

PLOS Mental Health

Journal Requirements:

1. Please send a completed 'Competing Interests' statement, including any COIs declared by your co-authors. If you have no competing interests to declare, please state "The authors have declared that no competing interests exist". Otherwise please declare all competing interests beginning with the statement "I have read the journal's policy and the authors of this manuscript have the following competing interests:"

Additional Editor Comments (if provided):

Reviewers' comments:

Reviewer's Responses to Questions

**Comments to the Author**

1. Does this manuscript meet PLOS Mental Health’s publication criteria? Is the manuscript technically sound, and do the data support the conclusions? The manuscript must describe methodologically and ethically rigorous research with conclusions that are appropriately drawn based on the data presented.

Reviewer #1: Yes

Reviewer #2: Yes

2. Has the statistical analysis been performed appropriately and rigorously?

Reviewer #1: Yes

Reviewer #2: Yes

3. Have the authors made all data underlying the findings in their manuscript fully available (please refer to the Data Availability Statement at the start of the manuscript PDF file)?

Reviewer #1: Yes

Reviewer #2: Yes

4. Is the manuscript presented in an intelligible fashion and written in standard English?

Reviewer #1: Yes

Reviewer #2: Yes

5. Review Comments to the Author

Reviewer #1: In this study, the authors utilize data from the Cross-sectional General Social Survey (GSS) collected between 2000 and 2022 to investigate attitudes toward suicide among individuals working in agriculture. Specifically, the authors explore the relationship between the attitudes toward suicide in this population, demographic factors, and spirituality.

Methodologically, the authors employ bivariate categorical statistical tests and multivariable logistic regression for their analysis. The primary hypothesis posited by the authors suggests that suicide is more accepted among those employed in agricultural occupations compared to the general population of the USA. However, the findings presented in the study do not support this hypothesis, adding complexity to the research.

This study contributes to the broader context of developing tailored approaches to address suicide prevention among farmworkers, a recognized at-risk group. The unexpected results prompt further inquiries into the underlying factors contributing to suicidal ideations within the agricultural population. Questions arise regarding the identification of specific risk and protective factors unique to this demographic and how this knowledge can inform improved suicide prevention strategies for agricultural workers.

While the authors present their findings effectively, there are minor issues that could enhance the clarity and rigor of the paper. Firstly, it is crucial to ensure that the sample of 167 agricultural worker respondents is nationally representative of the broader agricultural workforce in the United States. Additionally, providing more detailed information in the methodology section regarding the specific statistical methods employed would improve transparency and reproducibility. Lastly, sharing the psychometric properties of the GSS items used to measure suicide acceptability would aid in the interpretation of the findings, offering insights into the reliability of the measurement tools utilized.

I believe this paper makes a valuable contribution to the field by investigating the attitudes toward suicide among agricultural workers and advocating for tailored prevention strategies. With some minor revisions addressing the outlined issues, I recommend this paper for publication.

Minor issues:

1). The aim of the authors is to create a nationally representative study. The GSS dataset used by the authors is nationally representative of its 18,191 respondents, however it is not clear to me that the 167 respondents of the agricultural worker subgroup is a nationally representative of the agricultural workers of the United States.

2) The methodology section of the paper could benefit from more details. For example, the authors write “Bivariate categorical statistical tests (e.g., chi-square) of demographic variables and suicide

acceptability were used to examine differences between farm/ranch operators, farm/ranch

workers, and the general population.” (128-130), I would like to know which methods were used in which cases.

3) Sharing the psychometric properties of the GSS items that were used to measure suicide acceptability would aid the readers of this paper to interpret its findings. The authors support the use of these items by showing that they have been used in previous research, however we do not know whether the four items are reliable or not in measuring the acceptability of suicide.

Reviewer #2: It is one of the most crucial issues that our society will confront in the not-too-distant future, and I applaud the writers for going for it. When we're going through tough times, and I think this is a way to find meaning in life for the future when a human thinks others want to kill us (diseases, economic pressure or life course events).

The authors should present broad definitions in the opening part before digging into specific themes for instructional reasons. These definitions should cover factors such as occupation, demographics, and religiosity. There are a lot of people who are under the incorrect impression that these aspects are not significant and should simply be regarded control variables. Researchers are able to better comprehend the many influences and viewpoints that may have an effect on the results of the study by gaining a better understanding of these aspects, which are crucial since they give context. The writers are able to conduct a more thorough analysis of the data and arrive at results that are more accurate and that accurately reflect the diversity of the population that is being examined if they take into account aspects such as occupation, demographics, and religiosity. In the discussion section, there is potential for improvement regarding the issues of occupation, demographics, and religiosity as they pertain to the society of the United States of America. From the perspective of readers in the United States and all around the world, the conclusion needs to offer a mix of general and specific ideas that will assist in advancing this agenda in the course of future research. The factors of occupation, demographics, and religious affiliation all play a significant part in the formation of society dynamics and individual experiences in the framework of the United States. For example, one's occupation not only impacts their economic standing but also affects their access to resources, opportunities, and social networks. Age, ethnicity, and gender are all examples of demographic factors that lead to inequalities in education, healthcare, and representation. On the other hand, religiosity has an effect on social norms, values, and beliefs, which in turn has an effect on social cohesiveness and political ideologies. In order to achieve a full understanding of the society in the United States and to direct future study toward addressing issues that are relevant, it would be beneficial to recognize and analyze these elements in the discussion and conclusion sections. Furthermore, the interaction between work, demography, and religious affiliation can also be a contributor to social problems that bring individuals to the verge of suicide and cause them to consider ending their own lives. One example of a toxic environment that can contribute to feelings of hopelessness and despair is when a number of circumstances come together. These elements include unemployment, low socioeconomic position, discrimination based on demographics, and rigid religious beliefs. It is essential to have a solid understanding of these links in order to design effective interventions and support systems that can help prevent tragedies of this nature and improve mental well-being.

6. PLOS authors have the option to publish the peer review history of their article (what does this mean?). If published, this will include your full peer review and any attached files.

**Do you want your identity to be public for this peer review?** For information about this choice, including consent withdrawal, please see our Privacy Policy.

Reviewer #1: **Yes: **Aron B. Bekesi

Reviewer #2: No

---

## [Decision Letter · Decision Letter 1]

21 Oct 2024

PMEN-D-24-00103R1

Differences in Suicide Beliefs by Farming-Related Occupation, Demographic, and Religiosity Factors, General Social Survey 2000-2022

PLOS Mental Health

Dear Dr. Ward,

Thank you for submitting your manuscript to PLOS Mental Health. After careful consideration, we feel that it has merit but does not fully meet PLOS Mental Health’s publication criteria as it currently stands. Therefore, we invite you to submit a revised version of the manuscript that addresses the points raised during the review process.

We look forward to receiving your revised manuscript.

Kind regards,

Marc Eric Santos Reyes

Academic Editor

PLOS Mental Health

Journal Requirements:

Additional Editor Comments (if provided):

Reviewers' comments:

Reviewer's Responses to Questions

**Comments to the Author**

1. If the authors have adequately addressed your comments raised in a previous round of review and you feel that this manuscript is now acceptable for publication, you may indicate that here to bypass the “Comments to the Author” section, enter your conflict of interest statement in the “Confidential to Editor” section, and submit your "Accept" recommendation.

Reviewer #1: All comments have been addressed

Reviewer #3: (No Response)

2. Does this manuscript meet PLOS Mental Health’s publication criteria? Is the manuscript technically sound, and do the data support the conclusions? The manuscript must describe methodologically and ethically rigorous research with conclusions that are appropriately drawn based on the data presented.

Reviewer #1: Yes

Reviewer #3: Partly

3. Has the statistical analysis been performed appropriately and rigorously?

Reviewer #1: Yes

Reviewer #3: N/A

4. Have the authors made all data underlying the findings in their manuscript fully available (please refer to the Data Availability Statement at the start of the manuscript PDF file)?

Reviewer #1: Yes

Reviewer #3: (No Response)

5. Is the manuscript presented in an intelligible fashion and written in standard English?

Reviewer #1: Yes

Reviewer #3: Yes

6. Review Comments to the Author

Reviewer #1: Dear Dr. Ward and Dr. Blosnich,

I acknowledge the revisions you have made to the manuscript entitled "Differences in Suicide Beliefs by Farming-Related Occupation, Demographic, and Religiosity Factors, General Social Survey 2000-2022" (Manuscript Number: PMEN-D-24-00103R1).

I have carefully reviewed the changes made in response to the feedback provided. I appreciate the effort and thoroughness with which you have addressed the comments and suggestions. The revised manuscript is clear, well-organized, and provides valuable insights into the differences in suicide beliefs among various occupational, demographic, and religiosity groups, particularly in the context of the farming community.

The additional analyses and explanations the authors provided have significantly strengthened the manuscript. The clarity in presenting the findings and the comprehensive discussion on the implications of these beliefs for suicide prevention interventions are particularly commendable.

I believe the manuscript is now ready for publication in PLOS Mental Health. It offers a substantial contribution to the understanding of cultural and demographic factors influencing suicide beliefs and will be a valuable resource for researchers and practitioners in the field.

Thank you once again for your diligent work on this important study.

Sincerely,

Aron B Bekesi

Reviewer #3: Congratulations on this interesting study, as it elaborates on another construct or layer of experience about suicidality that may add to the understanding of suicide.

Introduction:

It is recommended that suicide beliefs should be discussed or defined and how this encompasses suicide acceptability or endorsement. Since the construct of suicide beliefs is what is in the title, kindly establish the scope and de/limitations. Are the terminologies suicide beliefs and suicide acceptability used interchangeably?

Methods:

Is the sample size of 167 people who worked in farming-/ranching-related representative of the population?

If the four items in General Social Surveys (GSS) “measure” suicide acceptability, do these four items “measure” suicide beliefs? Kindly use the term that is directly measured by your tool/questionnaire. Your scope and de/limitations may guide you to clearly phrase the title as well.

Additional details:

Please see additional details on the attached separate file for the recommendations

7. PLOS authors have the option to publish the peer review history of their article (what does this mean?). If published, this will include your full peer review and any attached files.

**Do you want your identity to be public for this peer review?** For information about this choice, including consent withdrawal, please see our Privacy Policy.

Reviewer #1: **Yes: **Aron B. Bekesi

Reviewer #3: No

---

## [Editor Report · Decision Letter 2]

4 Nov 2024

Differences in Suicide Acceptability by Farming-Related Occupation, Demographic, and Religiosity Factors, General Social Survey 2000-2022

PMEN-D-24-00103R2

Dear Dr. Ward,

We are pleased to inform you that your manuscript 'Differences in Suicide Acceptability by Farming-Related Occupation, Demographic, and Religiosity Factors, General Social Survey 2000-2022' has been provisionally accepted for publication in PLOS Mental Health.

Best regards,

Marc Eric Santos Reyes

Academic Editor

PLOS Mental Health